# Limitations of Active Learning with Deep Transformer Language Models

## Abstract

Active Learning (AL) has the potential to reduce labeling cost when training natural language processing models, but its effectiveness with the large pretrained transformer language models that power today's NLP is uncertain. We present experiments showing that when applied to modern pretrained models, active learning offers inconsistent and often poor performance. As in prior work, we find that AL sometimes selects harmful "unlearnable" collective outliers, but we discover that some failures have a different explanation: the examples AL selects *are* informative but also increase training instability, reducing average performance. Our findings suggest that for some datasets this instability can be mitigated by training multiple models and selecting the best on a validation set, which we show impacts relative AL performance comparably to the outlier-pruning technique from prior work while also increasing absolute performance. Our experiments span three pretrained models, ten datasets, and four active learning approaches.

## 1 Introduction

Deep learning models typically need large datasets of hand-labeled examples to perform well, but acquiring the labels can be expensive, often requiring human annotators to manually provide the label for every example. Active learning (AL) is a technique that aims to reduce this labeling cost by selecting only the most useful examples to receive labels (Settles, 2009).

AL has been shown to reduce labeling cost on a variety of NLP tasks in previous work (Shen et al., 2018; Siddhant & Lipton, 2018). However, these successes predate the rise of large pretrained Transformer language models (LMs) as a foundation for NLP. Whether AL's advantages extend to this setting is far less studied. Previous work has shown that AL consistently improves text classifiers that use pretrained Transformers (Lu & MacNamee, 2020; Ein-Dor et al., 2020), but those studies use a narrow range of model sizes and do not consider the most challenging semantic tasks such as question answering (QA) where pretrained LMs have made great strides in recent years. By contrast, Karamcheti et al. (2021) recently showed that AL is ineffective for large pretrained transformers in visual QA, and concluded that harmful collective outliers negatively impact AL and that heuristically pruning them improves the benefit of AL. However, whether these findings hold outside of visual QA is unknown.

In this paper, we present an expanded study of AL applied to fine-tuning large pretrained LMs. We investigate four AL approaches across ten datasets spanning text classification and recent multiple-choice question answering tasks. We find that AL fails to improve performance consistently, and often lowers performance significantly below that of a random-selection baseline. Further, our analysis suggests that in some cases, the failures have a different explanation than that offered by prior work. Specifically, AL sometimes select examples that are informative, but add *instability* to the training process, hurting average performance. As evidence, we show that a simple approach to reduce instability—training the model multiple times and selecting the best-performing on a held-out set, without any special treatment for collective outliers—improves the performance of AL by similar amount on average as the pruning in Karamcheti et al. (2021). Further, we find that the effectiveness of the pruning technique from Karamcheti et al. (2021) is often dataset-specific, improving the relative performance of AL over random sampling on some datasets and worsening it on others. In most cases, pruning also reduces the absolute performance across all methods, suggesting that it prunes helpful examples in addition to outliers.

To summarize, our contributions are:

1. We present experiments across four AL methods, ten data sets, and three models showing that AL has inconsistent performance with recent pretrained transformer LMs, often underperforming random selection especially as larger models are used.

2. We propose a novel explanation for some of AL's limitations: that it sometimes selects examples that are helpful but cause instability in training.

3. We conduct several ablation studies to rule out other factors and find that acquisition batch size and adversarial filtering do not meaningfully affect the performance of AL, while the amount of model pretraining has inconsistent effects between AL methods.

We hope our findings will provide guidance to other researchers considering applying AL to their data, and spur further investigation into more effective AL methods for pretrained LMs along with optimization techniques that are more robust to the challenging examples AL selects.

## 2 METHODS

### 2.1 TASK DEFINITION

Our active learning (AL) scenario is formulated as follows. We have a large pool of unlabeled data $\mathcal{U}$ and wish to select a subset $\mathcal{L}$ of size $Q$ to label that maximizes model performance on a held-out set. Given a model $\mathcal{M}$ and an AL scoring function $\mathcal{S}(x, \mathcal{M})$ (where $x$ is an unlabeled input), we iteratively select a batch of $|\Delta\mathcal{L}|$ examples from $\mathcal{U}$ according to $\mathcal{S}$, add these to $\mathcal{L}$, and retrain the model before selecting again. When re-training, we re-initialize all parameters to the original pretrained model state, following previous work (Ein-Dor et al., 2020; Hu et al., 2019). The first batch is selected randomly.

### 2.2 AL METHODS AND MODELS

We experiment with the following AL scoring functions:

- **Random** (the baseline);
- **Entropy**[1] of the model output distribution;
- **BALD-MC**, which is BALD (Houlsby et al., 2011) using Monte Carlo dropout (Gal & Ghahramani, 2016) to compute uncertainty;
- **BatchBALD-MC** (Kirsch et al., 2019), the batch variant of BALD;
- **Coreset** (Sener & Savarese, 2018), which selects examples that are distant from examples in $\mathcal{L}$ measured by distance between model output embeddings. We use the greedy variant of the algorithm and use the pre-trained model to pre-compute fixed output embeddings for the distance calculation.[2]

While many AL methods exist, we believe our choices form a representative set. Entropy is commonly used and was found to work well in the previous work on AL for Transformers (Lu & MacNamee, 2020). BALD has been found to outperform other methods in many cases, including NLP (Gal et al., 2017; Siddhant & Lipton, 2018). Coreset is diversity-based (rather than uncertainty-based like our other methods) and has outperformed other methods (including BALD) for deep models on image classification tasks (Sener & Savarese, 2018). We note that BALD and Entropy are originally online algorithms (selecting one example at a time) but are used in batch mode here. This is not theoretically sound but is commonplace and often effective (Kasai et al., 2019; Gal et al., 2017), so we include it for comparison. BatchBALD adds a diversity adjustment to BALD to account for this issue.

We test three pretrained Transformers, which are state-of-the-art models with different pretraining methods and sizes: **BERT-base** (110M parameters) (Devlin et al., 2019), **RoBERTa-large** (340M) (Liu et al., 2019), and **RoBERTa-base** (110M).

---

[1] A similar method is least-confidence (LC). We tried LC on most of our datasets and found it to be almost identical to entropy, so we omit it here.

[2] In preliminary experiments we tried training the model and updating the embeddings during acquisition, but this did not improve results and was more computationally expensive.

## 2.3 TASKS AND DATASETS

For datasets, we focus broadly on two tasks. The first is text classification, where AL has been effective with a limited class of Transformer models in previous work Lu & MacNamee (2020); Ein-Dor et al. (2020). This task allows us to verify our implementation against previous work, and to evaluate whether AL's effectiveness on it extends to a broader model class. The second is multiple-choice commonsense reasoning, a challenging NLP task where AL has not been evaluated, to our knowledge. The text classification datasets are AGN (Zhang et al., 2015), DBpedia (Zhang et al., 2015), Movie review sentence polarity (MRP) (Pang & Lee, 2005), and Movie review subjectivity (MRS) (Pang & Lee, 2004). The multiple-choice datasets are aNLI (Bhagavatula et al., 2020), CODAH (Chen et al., 2019), CommonsenseQA (CSQA) (Talmor et al., 2019), HellaSWAG (Zellers et al., 2019), PIQA (Bisk et al., 2020), and SWAG (Zellers et al., 2018). In the tables, "AGN-SB" and "DBpedia-SB" are AGN and DBpedia using only two classes (to create a binary classification task) and subsampled to have the same number of examples as reported by Lu & MacNamee (2020).

## 2.4 EXPERIMENTAL SETTINGS

We test on an eval set after each acquisition, which produces a learning curve of accuracy vs. data size. We report the area under the curve (AUC) as a measure of how effective the data selection strategy is.

We set $Q = 500$ and $|\Delta \mathcal{L}| = 25$ except where otherwise specified. The deep learning models were all trained by doing 1000 steps with the Adam optimizer and early stopping (hyperparameter details in Appendix A), and results are averaged over 10 trials.

## 3 RESULTS

We start by presenting our primary results showing that active learning offers inconsistent and often poor performance on our tasks. For RoBERTa-base, shown in Table 1, there are a few cases where AL outperforms the random-sampling baseline, but in many cases it gives no improvement (and in fact, for 10 of the 44 results it gives significantly *worse*). In the subsequent sections, we investigate several candidate explanations for the poor performance.

| Dataset \ Method | BALD-MC | BatchBALD-MC | Entropy | Coreset | Random |
|---|---|---|---|---|---|
| AGN-c | **86.8 (0.3)** | **87.5 (0.2)** | **87.3 (0.2)** | 86.3 (0.2) | 86.6 (0.2) |
| AGN-SB-c | **97.7 (0.0)** | **97.4 (0.0)** | **97.8 (0.0)** | **97.2 (0.0)** | 96.7 (0.1) |
| aNLI | 58.5 (0.2) | **58.9 (0.2)** | 57.8 (0.1) | **59.8 (0.1)** | 58.8 (0.2) |
| CODAH | 57.4 (0.5) | 58.5 (0.3) | 56.4 (0.3) | 57.6 (0.4) | 59.0 (0.6) |
| CSQA | 42.8 (0.3) | **43.9 (0.4)** | 43.3 (0.3) | **44.0 (0.2)** | 43.8 (0.2) |
| DBpedia-SB-c | **98.9 (0.0)** | **98.9 (0.0)** | **98.9 (0.0)** | **98.8 (0.0)** | 98.8 (0.1) |
| HellaSWAG | 38.1 (0.2) | 38.5 (0.2) | 38.1 (0.5) | **38.9 (0.2)** | 38.7 (0.2) |
| MRP-c | **83.4 (0.2)** | **83.3 (0.2)** | **83.5 (0.2)** | 81.8 (0.2) | 83.0 (0.3) |
| MRS-c | **94.9 (0.0)** | **95.3 (0.1)** | **95.2 (0.0)** | 94.0 (0.1) | 94.0 (0.2) |
| PIQA | 55.9 (0.3) | 55.9 (0.2) | 54.5 (0.4) | **57.1 (0.2)** | 56.8 (0.3) |
| SWAG | 62.5 (0.2) | 62.7 (0.2) | 60.5 (0.3) | 62.7 (0.2) | 63.5 (0.2) |
| Average | 70.64 (0.08) | **70.96 (0.07)** | 70.29 (0.08) | 70.75 (0.06) | 70.88 (0.08) |

Table 1: RoBERTa-base AUC results. In all tables, bold indicates better-than-random-sampling performance; gray background indicates significance at $p < 0.05$ in a two-tailed t-test. Standard errors are in parentheses. A "-c" suffix indicates text classification datasets.

## 3.1 COLLECTIVE OUTLIERS

Recently, Karamcheti et al. (2021) found that "collective outliers" can sometimes explain poor AL performance. Collective outliers are groups of points that together represent an out-of-distribution set of data that the model performs poorly on, and AL is prone to selecting these examples due to their difficulty. To investigate the extent to which this effect explains our results, we applied the same pruning strategy described in that work. This consists of creating a dataset map (Swayamdipta et al., 2020) which is made by first training the model for several epochs (10 for our experiments) on all

| Dataset \ Method | BALD-MC | BatchBALD-MC | Entropy | Coreset | Random |
|---|---|---|---|---|---|
| AGN-pruned50-c | **84.8 (0.3)** | **85.4 (0.3)** | **85.7 (0.4)** | **84.7 (0.3)** | 84.6 (0.3) |
| AGN-SB-pruned50-c | **97.7 (0.0)** | **97.5 (0.1)** | **97.6 (0.0)** | **97.2 (0.1)** | 96.9 (0.1) |
| aNLI-pruned50 | 57.1 (0.2) | 57.4 (0.1) | 52.7 (0.6) | **58.7 (0.2)** | 57.5 (0.2) |
| CODAH-pruned50 | **60.1 (0.6)** | 59.2 (0.4) | 58.1 (0.5) | 58.9 (0.3) | 58.4 (0.6) |
| CSQA-pruned50 | **46.5 (0.4)** | 46.0 (0.2) | **46.5 (0.3)** | 45.6 (0.3) | 46.1 (0.2) |
| DBPedia-pruned50-c | **99.1 (0.0)** | **99.1 (0.0)** | **99.0 (0.1)** | 99.0 (0.0) | 99.0 (0.0) |
| HellaSWAG-pruned50 | 35.0 (0.5) | 35.7 (0.5) | 35.6 (0.6) | **36.9 (0.3)** | 35.7 (0.3) |
| MRP-pruned50-c | **79.9 (0.4)** | **80.2 (0.3)** | **79.9 (0.3)** | 76.1 (0.4) | 79.0 (0.6) |
| MRS-pruned50-c | **94.4 (0.1)** | **94.9 (0.1)** | **94.8 (0.1)** | **94.1 (0.1)** | 94.1 (0.2) |
| PIQA-pruned50 | 56.0 (0.3) | 55.7 (0.3) | 56.0 (0.3) | **57.6 (0.1)** | 56.1 (0.3) |
| SWAG-pruned50 | 59.8 (0.3) | 59.4 (0.5) | 58.3 (0.3) | **59.9 (0.1)** | 59.8 (0.3) |
| Average | **70.04 (0.10)** | **70.06 (0.09)** | 69.48 (0.11) | **69.87 (0.07)** | 69.76 (0.10) |

Table 2: RoBERTa-base AUC results with 50% of the dataset pruned. Compared to the unpruned results, AL does better relative to random on average, but the improvement is not consistent across all datasets. A side-by-side comparison with Table 1 for BatchBALD can be found in Appendix C.

examples in the pool $\mathcal{U}$ and calculating the mean ("confidence") and variance ("variability") of the probability the model assigned to the correct label for that example during training. We then sort the examples by the product of confidence and variability and prune the lowest 50%[3]. Of course, this could not be done in a real-world scenario because it requires using the labels for the entire pool, but it is a useful way to gain insight into the behavior of AL.

Results are in Table 2, and show mixed results on the relative performance of AL vs random selection. The best AL method in our original experiments (BatchBALD-MC, Table 1) wins over random on the same number of data sets with pruning as without (7), and pruning only improves AL's relative gap with random on 6 of the 11 datasets. Other AL methods show qualitatively similar behavior. We also note that the change in absolute performance across the different datasets, with many datasets getting worse and only four (AGN-SB-c, CODAH, CSQA, and DBpedia-c) showing consistent improvements. This result suggests that collective outliers may not fully explain the poor performance of AL on some datasets, or at least that pruning based on confidence and variability is not always sufficient to isolate them.

## 3.2 MODEL SIZE AND INSTABILITY

| Dataset \ Method | BatchBALD-MC | Entropy | Coreset | Random |
|---|---|---|---|---|
| AGN-c | **88.7 (0.3)** | 87.3 (0.5) | 87.3 (0.1) | 88.0 (0.2) |
| aNLI | 64.0 (0.4) | 60.5 (0.4) | 64.5 (0.4) | 64.8 (0.3) |
| MRP-c | 80.0 (0.9) | 78.0 (0.8) | 84.3 (0.5) | 85.1 (0.6) |
| MRS-c | 93.5 (0.5) | 90.8 (0.7) | **94.9 (0.1)** | 94.9 (0.1) |
| SWAG | 68.0 (0.3) | 66.6 (0.3) | 69.3 (0.2) | 69.8 (0.4) |
| Average | 78.83 (0.24) | 76.65 (0.24) | 80.05 (0.13) | 80.53 (0.16) |

Table 3: RoBERTa-large AUC results. AL does substantially worse than random for most datasets. Due to the much greater computational cost of this model, BALD is omitted and only a subset of datasets are reported.

| Dataset \ Method | BatchBALD-MC | Entropy | Coreset | Random |
|---|---|---|---|---|
| AGN-c | **89.4 (0.1)** | **89.2 (0.1)** | 87.9 (0.1) | 88.6 (0.1) |
| aNLI | **68.2 (0.3)** | 65.8 (0.2) | **68.1 (0.2)** | 68.1 (0.3) |
| MRP-c | **87.2 (0.4)** | 87.0 (0.3) | 86.9 (0.3) | 87.1 (0.3) |
| MRS-c | **96.4 (0.1)** | **95.9 (0.2)** | **95.7 (0.1)** | 95.6 (0.1) |
| SWAG | 71.0 (0.3) | 69.1 (0.2) | 70.6 (0.1) | 71.8 (0.2) |
| Average | **82.44 (0.12)** | 81.41 (0.11) | 81.84 (0.08) | 82.23 (0.10) |

Table 4: RoBERTa-large AUC results using the convex hull of each learning curve. AL does much better here compared to Table 3, suggesting that training instability was a large factor in its previous failures.

---

[3]An ablation with 25% pruning can be found in Appendix C

Compared to most previous studies where AL has been effective for language tasks, the models we use have many more trainable parameters. To test whether model size may be a contributor to AL's poor performance, we evaluate the much larger RoBERTa-large (3x more parameters than RoBERTa-base) in Table 3, but limit this experiment to a subset of datasets due to the high computational cost of the model.

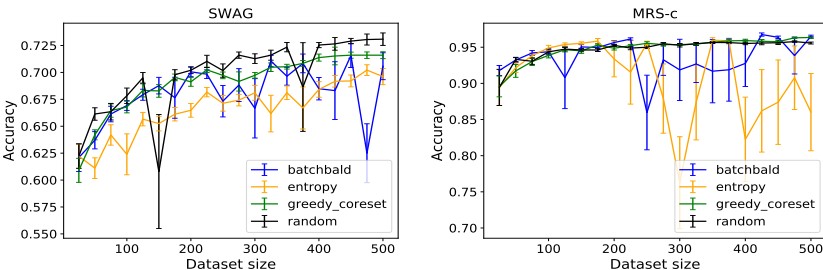

Figure 1: Averaged learning curves for SWAG and MRS with RoBERTa-large. Bars represent standard error. Best viewed in color.

The results show that AL is substantially less effective for the larger model. In fact, for BatchBALD and Entropy, RoBERTa-large shows more negative impact from AL than RoBERTa-base across all five datasets we evaluate in this experiment. The average difference between BatchBALD and Random is -1.7 for RoBERTa-large but +0.4 for RoBERTa-base (for the same datasets). One reason for this is an increased frequency of degenerate training runs for RoBERTa-large when using AL, which have unusually low accuracy (often no better than guessing) and manifest in Figure 1 as points that dip sharply below the global trend of the learning curve and have high variance. For example, most results in the second half of the learning curve for SWAG are above 68% accuracy, but with AL several runs randomly drop to anywhere from 25-67%.

Such degenerate cases with large transformers are well-known (Devlin et al., 2019). But the substantial increase in failed training runs when using AL leads us to believe that AL methods select sets of examples that amplify the *instability* of the optimization process, which we define here as the variance in test set accuracy observed when training multiple models (with different random seeds) on the same set of data. The increased instability means that the selected examples are not necessarily harmful as the collective outlier theory suggests (because not all training runs fail), but instead are potentially useful examples that are hard to learn from effectively.

Correcting for these failed training runs is hard for two reasons: (1) they are frequent enough that it would be computationally prohibitive for us to re-run all of them with different random seeds until getting a success, as in previous work (Devlin et al., 2019; Porada et al., 2019), and (2) the model does not always degrade to random-label performance, meaning a threshold would need to be chosen for what consitutes a "failure", and this choice may affect the conclusions.

To roughly approximate how well the model could do if it did not have these optimization failures, we consider the area under the convex hull of the learning curves instead of including all points, and display the results in Table 4. This substantially improves AL relative to random and suggests that the increased frequency of failed training runs is the main culprit for the poor performance of AL. Because the learning curve of accuracy vs dataset size is generally convex, our convex hull approximation is an underestimate of what we would expect if we actually achieved the best optimization that the model could reasonably achieve with the selected data.

To a lesser extent, we find evidence of AL-induced instability on RoBERTa-base as well. Figure 2 shows some of the learning curves from the Table 1 results, and while they are much less noisy than the RoBERTa-large table, it still qualitatively appears that AL has more noise than random on the datasets where AL loses. Quantitatively, we compute the average difference between the convex hull AUC and the normal AUC, which serves to estimate the level of instability in the absence of repeated trials on the same data. On the datasets where AL loses, BatchBALD's average difference (2.04) is larger than Random's (1.80), whereas on datasets where AL wins the average differences are equal (0.77). That is, AL shows greater instability on the losses than on the wins, suggesting that instability may play a role in the failures.

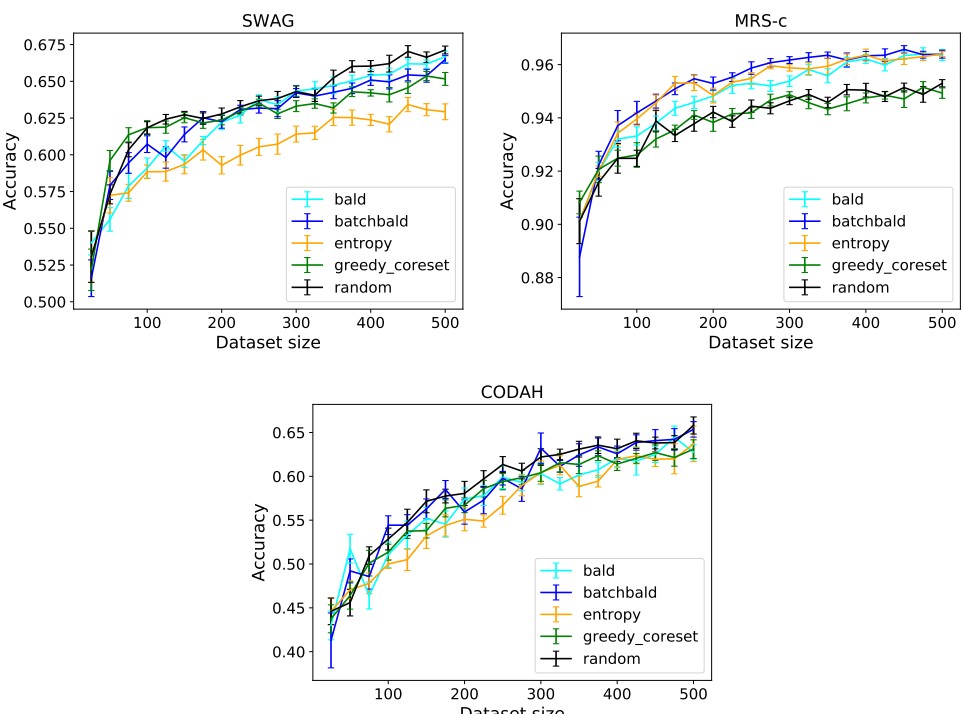

Figure 2: Averaged learning curves for SWAG, MRS-c and CODAH with RoBERTa-base. Bars represent standard error. Best viewed in color.

### 3.3 Multi-training RoBERTa-base

| Dataset \ Method | Multi-training | | Original (from Table 1) | |
|---|---|---|---|---|
| | BatchBALD-MC | Random | BatchBALD-MC | Random |
| AGN-c | **87.5 (0.3)** | 87.0 (0.1) | **87.5 (0.2)** | 86.6 (0.2) |
| aNLI | 59.5 (0.2) | 59.7 (0.1) | **58.9 (0.2)** | 58.8 (0.2) |
| CODAH | **59.0 (0.8)** | 58.7 (0.4) | 58.5 (0.3) | 59.0 (0.6) |
| CSQA | 44.6 (0.3) | 45.2 (0.2) | **43.9 (0.4)** | 43.8 (0.2) |
| DBpedia-SB-c | **98.9 (0.0)** | 98.8 (0.0) | **98.9 (0.0)** | 98.8 (0.1) |
| HellaSWAG | **40.1 (0.1)** | 39.7 (0.2) | 38.5 (0.2) | 38.7 (0.2) |
| MRP-c | **84.5 (0.1)** | 83.7 (0.2) | **83.3 (0.2)** | 83.0 (0.3) |
| MRS-c | **95.3 (0.1)** | 94.2 (0.1) | **95.3 (0.1)** | 94.0 (0.2) |
| PIQA | 56.7 (0.2) | 57.1 (0.3) | 55.9 (0.2) | 56.8 (0.3) |
| SWAG | 63.8 (0.3) | 64.1 (0.2) | 62.7 (0.2) | 63.5 (0.2) |
| Average | **68.99 (0.10)** | 68.84 (0.07) | **68.32 (0.07)** | 68.30 (0.09) |

Table 5: Multi-trained RoBERTa-base AUC results with 5 trainings per point on the learning curve. Original results from Table 1 are reprinted for comparison. Absolute numbers are significantly better with multi-training but relative results are mixed.

Because RoBERTa-base is several times smaller than RoBERTa-large, it is feasible to do a more thorough investigation of the impact of instability by training multiple times on each collected dataset along each learning curve to ensure that we get a well-optimized model ("multi-training"). We repeat our experiments with BatchBALD and random sampling, training the model 5 times after each batch acquisition and picking the best on the dev set to use both for evaluation and for selecting the next batch from the pool. Results are reported in Table 5.

We find that multi-training improves BatchBALD relative to random on all of the datasets where BatchBALD underperformed random in Table 1. However, the improvements are small enough to attribute to noise in some cases, and in some of the cases where BatchBALD previously outperformed

random, it became relatively worse after multi-training. We conclude from this that the effect of instability on smaller models is less pronounced than on larger models.

Finally, we note that the effects of multi-training are similar to those of pruning in Table 2; for example, both substantially improve AL on CODAH and slightly worsen it on CSQA. However, multi-training improves the absolute performance across almost all results ($p < 10^{-4}$ for a related-sample t-test), whereas pruning mostly reduces it ($p < 0.02$). Multi-training should improve accuracy if instability is an issue, but there is no reason to believe it helps the model to ignore collective outliers in the training set. Thus, the fact that a similar improvement to pruning can be achieved by multi-training without pruning suggests that instability is a distinct (but complementary) phenomenon from collective outliers.

## 3.4 OTHER ABLATIONS

Although instability appears to largely explain poor AL performance on large models and collective outliers explain it on some datasets, there are several datasets for which AL still fails to outperform random on RoBERTa-base even after attempting to correct for these effects. In this section, we investigate a variety of alternative hypotheses for the results we observed.

### 3.4.1 QUESTION FORMAT

| Dataset \ Method | Entropy | Coreset | Random |
|---|---|---|---|
| **RoBERTa (base)** | | | |
| aNLI-c | 50.9 (0.2) | 51.8 (0.1) | 53.1 (0.1) |
| CODAH-c | 59.9 (0.5) | 60.0 (0.2) | 60.3 (0.5) |
| HellaSWAG-c | 51.2 (0.2) | **54.0 (0.1)** | 51.8 (0.2) |
| PIQA-c | 50.3 (0.1) | **50.6 (0.1)** | 50.5 (0.1) |
| SWAG-c | 63.9 (0.4) | **65.0 (0.1)** | 64.3 (0.6) |
| Average | 55.22 (0.13) | **56.29 (0.06)** | 55.98 (0.15) |
| **BERT (base)** | | | |
| aNLI-c | 50.3 (0.1) | 50.3 (0.1) | 50.8 (0.1) |
| SWAG-c | 59.3 (0.4) | **60.4 (0.2)** | 59.5 (0.3) |
| Average | 54.81 (0.20) | **55.38 (0.12)** | 55.15 (0.17) |

Table 6: AUC results for classification versions of multiple-choice datasets. Compared to Table 1, AL relatively improves on some datasets and worsens on others vs random, with no clear pattern. BALD and BatchBALD are omitted here due to their high cost.

In Table 1, we see that AL typically performs better on the text classification datasets than the commonsense reasoning ones. One difference between these tasks is the kind of language competency they attempt to evaluate, but another is simply their format: the text classification datasets require choosing one of a fixed set of (typically two) labels, whereas the multiple-choice tasks require selecting which of multiple (question-specific) passages is correct. We investigate this confound by creating text classification versions of the multiple-choice datasets, where each text classification example is the multiple-choice prompt concatenated with one answer choice and the classification task is to determine if the given answer is correct. For multiple-choice tasks with more than two answer choices, we subsample the resulting dataset to balance the number of positive and negative examples.

The results, displayed in Table 6, do not show a consistent effect of question format on AL. SWAG and CODAH in the text classification format appear to get a bigger boost from AL than in the multiple-choice format, but aNLI does slightly worse and HellaSWAG and PIQA stay roughly the same. We conclude that the trend observed in Table 1 may have more to do with the typical types of reasoning used by classification vs. multiple-choice datasets than with the format used to present them to the model.

### 3.4.2 ADVERSARIAL FILTERING

A possible explanation for the failure of AL on commonsense reasoning tasks compared to text classification is that many of the multiple-choice datasets are adversarially filtered Bras et al. (2020). This means questions that a baseline model gets right in cross-validation are removed during dataset

construction, which may have a similar effect to AL and cancel out the benefit. The adversarially-filtered datasets are aNLI, SWAG, CODAH, PIQA, and HellaSWAG, which together make up the majority of cases where AL fails in our experiments.

To probe the effect of filtering, we obtained the unfiltered version of aNLI from its authors. The Entropy, Coreset, and Random methods resulted in AUC values of 78.3 (0.2), 79.8 (0.3), and 80.8 (0.2) respectively (due to the large size, running BALD was not feasible). Perhaps surprisingly, this is no better than the result with the filtered dataset, suggesting that adversarial filtering does not explain the poor performance of AL.

### 3.4.3 PRETRAINING

| Dataset \ Method | BatchBALD-MC | Entropy | Coreset | Random |
|---|---|---|---|---|
| AGN-c | **87.8 (0.2)** | 87.3 (0.2) | 87.1 (0.2) | 87.3 (0.2) |
| AGN-SB-c | **97.8 (0.1)** | **98.0 (0.0)** | **97.5 (0.0)** | 97.1 (0.1) |
| aNLI | **52.9 (0.1)** | 52.3 (0.1) | 52.3 (0.1) | 52.8 (0.1) |
| CODAH | **48.3 (0.4)** | 44.8 (0.6) | 47.7 (0.6) | 48.1 (0.7) |
| CSQA | **29.1 (0.4)** | 28.4 (0.3) | 27.6 (0.2) | 28.5 (0.2) |
| DBpedia-SB-c | **99.1 (0.0)** | **99.1 (0.0)** | **99.1 (0.0)** | 99.0 (0.0) |
| HellaSWAG | **29.9 (0.3)** | 29.0 (0.4) | **30.4 (0.3)** | 29.8 (0.3) |
| MRP-c | **80.5 (0.2)** | 79.7 (0.3) | 77.5 (0.4) | 80.3 (0.4) |
| MRS-c | **94.5 (0.1)** | **94.3 (0.1)** | 92.5 (0.1) | 93.2 (0.1) |
| PIQA | **52.6 (0.3)** | **52.6 (0.2)** | **52.6 (0.2)** | 52.3 (0.2) |
| SWAG | **60.6 (0.4)** | 56.0 (0.4) | 58.2 (0.4) | 60.4 (0.2) |
| Average | **66.63 (0.08)** | 65.58 (0.09) | 65.70 (0.08) | 66.26 (0.09) |

Table 7: BERT-base AUC results.

Previous studies have found that model regularization (Munjal et al., 2020) and semi-supervised training (Chan et al., 2021) can reduce the effectiveness of AL, so we investigate whether unsupervised pre-training may have a similar effect. Using a randomly-initialized model unfortunately produces random-baseline performance in all cases, so we instead use BERT-base for comparison. Because BERT and RoBERTa have the same architecture but RoBERTa uses much more pretraining data (roughly 10x), we expect that if pretraining reduces AL effectiveness then RoBERTa should benefit less from AL than BERT. The results for BERT are reported in Table 7, and by comparing them to the RoBERTa results in Table 1 we see that BERT generally gets better relative performance with BatchBALD, worse with Coreset, and about the same with Entropy. This suggests that the effect of pretraining varies between methods. We suspect that the reason Coreset does better with RoBERTa may be related to better pretraining producing better embeddings with which to construct the core-set. We also note that the benefit from BatchBALD on any individual dataset is generally small and not significant, but due to its consistency across datasets we find a significant ($p < 0.008$) difference in the average performance vs Random in a related-sample t-test.

### 3.4.4 ANNOTATION BATCH SIZE

As an additional ablation, we consider the possibility that the annotation batch size $|\Delta\mathcal{L}|$ is critical to the success of AL, and the necessary batch size is simply too small to be feasible for large models, because computation cost is inversely proportional to the batch size. To test this, we run experiments with batch size 12 instead of 25 for the AGN and SWAG datasets in Table 8, and find that using a batch size of 12 produces a similar result to a batch size of 25.

| Dataset \ Method | Entropy | Random |
|---|---|---|
| AGN-c | **87.3 (0.3)** | 86.1 (0.2) |
| SWAG | 59.7 (0.2) | 62.6 (0.2) |

Table 8: RoBERTa-base AUC results with $|\Delta\mathcal{L}| = 12$. Overall results are slightly lower than in Table 1, likely due to the extra point on the learning curve where dataset size is 12, but the differences between AL and random are roughly unchanged.

## 4 RELATED WORK

Many AL methods have been proposed. Many use some estimate of model uncertainty as a heuristic for useful examples; this includes model ensemble agreement (Prabhu et al., 2019), Monte Carlo dropout (Gal & Ghahramani, 2016), and a variety of simpler methods such as entropy of the model output (Settles, 2009). Others have found that encouraging diversity can be more robust, particularly in the context of deep learning (Sener & Savarese, 2018; Sinha et al., 2019). AL has been found to be effective in multiple domains, including computer vision (Gal et al., 2017) and natural language processing (Siddhant & Lipton, 2018).

Despite its successes, recent work has found several cases where the benefit of AL seems to disappear. Munjal et al. (2020) finds that model regularization overlaps with the gain from AL, with better-regularized models having higher task performance but receiving no benefit from AL. Several studies (Mittal et al., 2019; Chan et al., 2021; Siméoni et al., 2019) have found that self-supervised and semi-supervised learning can have a similar effect, improving model performance but failing to stack with boosts from AL. Lowell et al. (2019) finds that the benefits of AL on text classification tasks often do not transfer between models.

Unlike the work mentioned above, our focus is on evaluating AL with large pretrained Transformer language models. Previous results have shown that AL *can* be effective for certain pretrained transformers on text classification tasks Lu & MacNamee (2020); Ein-Dor et al. (2020). Our results qualitatively agree with the prior work when we evaluate on the same datasets and models, but we show how AL becomes far less effective when moving to larger and better-performing pretrained models or richer commonsense reasoning tasks.

Similar to our results, Karamcheti et al. (2021) recently found a negative result for AL in the domain of visual question-answering, and discovered that the effect was largely explained by "unlearnable" collective outliers that the model often fails on even during training. However, we find that applying their pruning technique to our ten datasets and using recent batch-aware AL methods (Kirsch et al., 2019) yields mixed results, suggesting that the impact of collective outliers may vary by dataset.

Finally, with regard to the instability of large transformers, Liu et al. (2020) find that the position of layernorm relative to residual connections in transformer layers can cause instability by amplifying the effects of small parameter perturbations. This may be related to the effects we observe, as in some sense AL selects examples that maximally change the model, making it more susceptible to this effect. Further investigating this and applying the proposed Admin initialization technique to resolve the instability is an interesting avenue for future work.

## 5 CONCLUSION

We have investigated active learning (AL) with large pretrained Transformers on a variety of tasks not covered by previous experiments, finding that AL offers poor and inconsistent performance in many cases. We identify a novel explanation for some of these failures in the case of large models: an increase in instability, as evidenced by e.g. a much higher frequency of cases where RoBERTa-large fails to converge. Attempting to resolve the instability by selecting the best of multiple trainings on each selected dataset behaves similarly to an outlier-pruning technique from prior work in terms of relative performance between AL and random, but with higher absolute performance. This suggests that multiple trainings may be a preferable approach for studying AL compared to pruning, as it allows measurements in a more practically relevant (higher-performing) setting. Finally, a variety of other ablations suggest that factors such as acquisition batch size, adversarial filtering, and input format are not responsible for the suboptimal behavior of AL, and the effects of greater model pretraining are inconsistent between AL methods.

Several issues remain to be addressed in future work. Some experiments are limited due to high computational cost, so a more exhaustive investigation of whether further improvements in model optimization would enable AL to outperform random in all cases remains a task for future work. In addition, tests with a wider range of models and model sizes would be valuable for confirming the generality of these findings. Lastly, we hope our findings will enable the development of improved active learning and optimization algorithms that do not suffer from the drawbacks we discovered.

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

## A    Hyperparameters

| Parameter | Value |
|---|---|
| Learning rate | 2e-5 |
| Adam warmup steps | 150 |
| Batch size | 16 |
| Adam $\epsilon$ | 1e-6 |
| Adam $\beta_1$ | 0.9 |
| Adam $\beta_2$ | 0.98 |
| Weight decay | 0.01 |

Table 9: Transformer hyperparameters

Transformer models were fine-tuned for a fixed 1,000 steps, evaluating on the dev set every 100 steps and using the checkpoint with best dev accuracy to obtain the final test set accuracy. We found from preliminary experiments that the hyperparameters in Table 9 generally worked well on all models and datasets. The learning rate followed a sloped triangular schedule, warming up over the first several steps and then linearly decreasing until the end of training. The maximum sequence length for models was set per-dataset based on the length of input texts. In general it was set so that at least 99% of examples fit completely within the sequence length, and those that did not fit were pruned.

For BALD, the number of Monte Carlo dropout samples was set to 5, and we found in preliminary experiments on CSQA and MRP that increasing the number of samples to 20 did not substantially improve results. We performed the Monte Carlo dropout using the normal dropout probabilities in the dropout layers of the models (generally 0.1); increasing the dropout probabilities to 0.5 (just for the BALD step, not model training) added noise due to the large number of dropout layers and actually caused performance to slightly decrease in our preliminary tests.

The experiments were run on servers with Nvidia RTX 2080 Ti and GTX 1080 gpus. We estimate that all reported experiments together represent about 3,000 gpu-hours for an RTX 2080 Ti.

Our experiments are implemented using Pytorch (Paszke et al., 2019) and code is available in the supplementary material. We used version 2.4.1 of the HuggingFace Transformers library (Wolf et al., 2020), which has a small bug in the RoBERTa implementation. Testing two of our datasets with a newer version suggests that this doesn't change our conclusions.

## B    Dataset sizes

See Table 10.

## C    Pruning Level

For our main experiments we pruned the 50% of examples most likely to be collective outliers, as that threshold is what appeared to work best in the work of Karamcheti et al. (2021). However, as we did not find pruning to be as effective as in that work, we also tried a 25% threshold with just one AL method (BatchBALD) to see if the pruning threshold made a difference. The results are displayed in Table 11. We observe that both the relative gain from AL and the absolute performance fall between the corresponding numbers for 0% and 50% on average, and therefore conclude that while the pruning threshold could be adjusted to trade relative and absolute gains, it would not produce a boost in both at once across our datasets.

## D    Label Budget

We primarily focus on a low-budget setting in this work, but it is interesting to consider how performance might improve with more labels. In Table 12 we show the results of experiments with

| Dataset | Train | Dev | Test |
|---------|-------|-----|------|
| AGN-SB-c | 2000 | 2500 | 5000 |
| DBpedia-SB-c | 2000 | 2000 | 2000 |
| CODAH | 2376 | 100 | 200 |
| CODAH-c | 4752 | 200 | 400 |
| CSQA | 9141 | 300 | 1221 |
| MRP-c | 8614 | 512 | 1024 |
| MRS-c | 9000 | 500 | 500 |
| PIQA | 14113 | 1000 | 1838 |
| PIQA-c | 28226 | 2000 | 3676 |
| HellaSWAG | 39905 | 1000 | 8042 |
| HellaSWAG-c | 65198 | 1988 | 15508 |
| SWAG | 65354 | 4096 | 20006 |
| AGN-c | 100000 | 5000 | 10000 |
| SWAG-c | 130708 | 8192 | 40012 |
| aNLI | 150000 | 1036 | 1532 |
| aNLI-c | 300000 | 2072 | 3064 |

Table 10: Dataset sizes. "Train" is the set we used as the unlabeled pool for active learning. Note also that the numbers add up to slightly less than the official sizes of these datasets, as we held out some additional data that ultimately went unused in this work.

| Dataset \ Method | 0% pruned (from Table 1) | | 25% pruned | | 50% pruned (from Table 2) | |
|---|---|---|---|---|---|---|
| | BatchBALD-MC | Random | BatchBALD-MC | Random | BatchBALD-MC | Random |
| AGN-c | **87.5 (0.2)** | 86.6 (0.2) | **87.0 (0.2)** | 86.6 (0.2) | **85.4 (0.3)** | 84.6 (0.3) |
| AGN-SB-c | **97.4 (0.0)** | 96.7 (0.1) | **97.4 (0.1)** | 96.8 (0.1) | **97.5 (0.1)** | 96.9 (0.1) |
| aNLI | **58.9 (0.2)** | 58.8 (0.2) | 58.1 (0.1) | 58.4 (0.1) | 57.4 (0.1) | 57.5 (0.2) |
| CODAH | 58.5 (0.3) | 59.0 (0.6) | 58.9 (0.5) | 60.2 (0.5) | **59.2 (0.4)** | 58.4 (0.6) |
| CSQA | **43.9 (0.4)** | 43.8 (0.2) | **45.8 (0.3)** | 45.2 (0.3) | 46.0 (0.2) | 46.1 (0.2) |
| DBPedia-c | **98.9 (0.0)** | 98.8 (0.1) | **99.1 (0.0)** | 99.0 (0.1) | **99.1 (0.0)** | 99.0 (0.0) |
| HellaSWAG | 38.5 (0.2) | 38.7 (0.2) | **37.4 (0.4)** | 37.2 (0.4) | 35.7 (0.5) | 35.7 (0.3) |
| MRP-c | **83.3 (0.2)** | 83.0 (0.3) | **82.6 (0.2)** | 81.8 (0.2) | **80.2 (0.3)** | 79.0 (0.6) |
| MRS-c | **95.3 (0.1)** | 94.0 (0.2) | **95.0 (0.1)** | 94.0 (0.1) | **94.9 (0.1)** | 94.1 (0.2) |
| PIQA | 55.9 (0.2) | 56.8 (0.3) | 55.8 (0.3) | 56.2 (0.2) | 55.7 (0.3) | 56.1 (0.3) |
| SWAG | 62.7 (0.2) | 63.5 (0.2) | **61.1 (0.2)** | 60.7 (0.3) | 59.4 (0.5) | 59.8 (0.3) |
| Average | **70.96 (0.07)** | 70.88 (0.08) | **70.75 (0.08)** | 70.54 (0.08) | **70.06 (0.09)** | 69.76 (0.10) |

Table 11: Roberta-base with different levels of pruning.

| Dataset \ Method | $Q$=1000, $|\Delta L|$=50 | | $Q$=500, $|\Delta L|$=25 (from Table 1) | |
|---|---|---|---|---|
| | BatchBALD-MC | Random | BatchBALD-MC | Random |
| AGN-c | **88.6 (0.1)** | 88.2 (0.1) | **87.5 (0.2)** | 86.6 (0.2) |
| AGN-SB-c | **97.7 (0.1)** | 97.3 (0.1) | **97.4 (0.0)** | 96.7 (0.1) |
| aNLI | 60.4 (0.2) | 60.5 (0.2) | **58.9 (0.2)** | 58.8 (0.2) |
| CODAH | **62.6 (0.4)** | 62.1 (0.4) | 58.5 (0.3) | 59.0 (0.6) |
| CSQA | **47.8 (0.4)** | 47.7 (0.2) | **43.9 (0.4)** | 43.8 (0.2) |
| DBpedia-SB-c | **99.1 (0.0)** | 99.1 (0.0) | **98.9 (0.0)** | 98.8 (0.1) |
| HellaSWAG | 41.2 (0.1) | 41.2 (0.1) | 38.5 (0.2) | 38.7 (0.2) |
| MRP-c | **85.6 (0.1)** | 84.5 (0.3) | **83.3 (0.2)** | 83.0 (0.3) |
| MRS-c | **95.8 (0.1)** | 95.2 (0.0) | **95.3 (0.1)** | 94.0 (0.2) |
| PIQA | **57.7 (0.1)** | 57.4 (0.3) | 55.9 (0.2) | 56.8 (0.3) |
| SWAG | 65.7 (0.3) | 65.8 (0.2) | 62.7 (0.2) | 63.5 (0.2) |
| Average | **72.92 (0.06)** | 72.63 (0.07) | **70.96 (0.07)** | 70.88 (0.08) |

Table 12: RoBERTa-base AUC results with $Q = 1000$ and $|\Delta L| = 50$. Original results from Table 1 are reprinted for comparison. AL has higher relative performance on the results with greater label budget and batch size.

budget $Q$=1000 and batch size[4] $|\Delta L|$=50. The results show that a higher budget generally results in better relative AL performance. This result is encouraging, as it suggests the instability of AL-selected datasets is mitigated simply by collecting more labels. Of course, it does not help in cases where the labels are costly and the budget cannot easily be increased, so an interesting question for future work is how to determine the minimum viable label budget needed for AL to become effective.

---

[4]Unfortunately we cannot avoid changing multiple variables here, as either the batch size or the number of batches must increase when we use a larger budget. Due to our earlier batch size ablation we consider it unlikely that the batch size plays a large role, and generally attribute the results here to the budget increase.

