# OpenReview forum: "Limitations of Active Learning With Deep Transformer Language Models"
_ICLR.cc/2022/Conference — ICLR 2022 Submitted_

### Official Review · Reviewer_DPi3 · 2021-10-28

**Correctness:** 3
**Technical Novelty And Significance:** 2
**Empirical Novelty And Significance:** 3
**Recommendation:** 6
**Confidence:** 3

**Main Review:**

Strengths:

The topic investigated in this work is interesting and would be useful for further works since pre-trained models have been shown effective at learning with small amounts of annotations and AL is also a method with similar spirit.

This paper provides a good and solid empirical analysis on this topic with various different models, datasets and AL methods.

There are several detailed ablation analyses which I find interesting, such as the influence of question format and adversarial filtering on the datasets.

Weakness:

One of my major concerns is on the point of training instability. I feel that this might not be the problem of AL, but at least some part of the problem is that we need a more robust way to train the model? Surely the random baseline is also treated in similar ways, so the comparisons are fair. Especially, fine-tuning might be sensitive to hyper-parameters, and it seems that there are no specific tunings? Surely I understand the cost of doing this, but I think it would be nice to select some of the datasets to carry out some hyper-parameter tuning.

As discussed in Section 3.4.4, annotation batch size could be important. I think there should be more experiments on this, for example, with larger sizes rather than smaller.

One specific question that I have but might not be fully answered in this work is that: why are the examples selected by AL more likely to cause the training instability? Some experiment results show that outliers might not tell the full story. Then why does this happen, is it related with instance diversity or even simpler label diversity (for example, maybe in some iterations, AL only chooses instances with the same label)? I'm quite interested to see more analysis on this.

Other comments:

For the hyper-parameter issue above, what are the differences of multiple runs in the multi-training (in Section 3.3), only changing seed or changing hyper-parameters? If only seed, I suggest doing similar experiments with some degree of tuning of hyper-parameters.

It would be nice to have results where all the datasets are available as in standard supervised settings as upper-bounds.

It would be interesting to explore tasks beyond simple classification.


**Summary Of The Paper:**

This paper investigates the usefulness of Active Learning (AL) on a variety of NLP tasks with pre-trained models. The authors show that with pre-trained models, AL can sometimes offer inconsistent and even poorer results than random baselines. In addition to outliers, they show that training instability is a major issue. There are also various further ablation studies on this issue.


**Summary Of The Review:**

Generally I find that this work provides a reasonable study on the usefulness of AL with pre-trained models, there are lots of experiments and there are certain interesting findings. Nevertheless, I still have concerns with the training setup and there are still some analyses that I would like to see.

---

> ### Author Response · Authors · 2021-11-23
> **Response to reviewer 5**
>
> Thank you for your review.
>
> We believe that both optimization and example selection together play a role in instability, although in this work we focus on the latter.  In limited preliminary experiments, neither learning rate nor batch size appeared to have large effects.  A proper hyperparameter ablation with a small number of datasets and settings would take additional time, but we will add this to the final version of our paper.
>
> Regarding multi-training: We only change the seed, which controls example order, weight initialization, and dropout masks.
>
> Your question about the kinds of selections AL makes is similar to a question asked by Reviewer 1, so we provide the same response here:
>
> From our exploratory analysis we have observed that Coreset heavily favors longer examples, and generally Entropy does as well to a far lesser extent. Analysis of other qualitative patterns among selected examples is still an item for future work, as thus far we have not observed any clear trend.
>
> On the specific point of label diversity, we did observe some bias with AL (e.g., 55/45 for MRP, 30/27/26/16 for AGN), but we don't believe this could explain our observations because the multiple-choice tasks cannot have such label bias, as the answer choices are given to the model in an order-independent way (the same way the original BERT paper used for SWAG).

---

> > ### Comment · Reviewer_DPi3 · 2021-11-29
> > **Response to authors**
> >
> > Thank you for your response, which roughly answers my questions. It will be nice to have some extra results on the hyper-parameter ablation. At the same time, I also suggest to investigate into more details of the cases of degenerated training, especially the cases where performances suddenly drop (for example points shown in Figure 1) and there might be some patterns.

---

### Official Review · Reviewer_Seyb · 2021-10-31

**Correctness:** 3
**Technical Novelty And Significance:** 3
**Empirical Novelty And Significance:** 2
**Recommendation:** 5
**Confidence:** 3

**Main Review:**

Strengths: The experimental scale of this work is undoubtedly large and comprehensive, involving many data sets, different types of active learning algorithms. It is constructive for the researchers who tried to use active learning methods for reducing the labeling cost in NLP tasks.

Weaknesses:
1) compare with [1], the research questions and the corresponding conclusions of this work are not enough novel and attractive. Actually, current studies have involved: i) marginal benefits of deep active learning methods; ii) employing active learning with pre-trained LMs like Bert on regular NLP tasks; iii) employing active learning on challenging tasks like VQA. In fact, these studies have covered almost all existing research topics that worth studying and revealed the problems existing in deep active learning. For now, it is more meaningful to address these issues than to continue to discuss whether deep active learning has similar problems in some subdivided tasks.

2) In the experiments, the authors only adopt 500 quota for each AL experiment, it's too small. For example, in Figure 2 (CODAH), the performance of AL methods are far from convergence, the author should adopt more quota or at least provide the performance trained on full dataset, like [2].
--------------------------------------- Minor Issues -------------------------------------------

3) Many writing problems could be found in this paper. For instance, in page 2, in contribution 2, the sentence is incompleted. In section 2.2, should add ';' or '.' at the end of each item when describing the AL algorithms.

4) It is inaesthetic to plot standard error bar, like Figure 1, can use plt.fill_between instead of plt.errorbar.
5) Should pay more attention to design the layout of tables, e.g., in Table 6, it should be Method | Dataset | AL methods.


References:
[1] Karamcheti S, Krishna R, Fei-Fei L, et al. Mind your outliers! investigating the negative impact of outliers on active learning for visual question answering[J]. arXiv preprint arXiv:2107.02331, 2021.

[2] Margatina K, Barrault L, Aletras N. Bayesian Active Learning with Pretrained Language Models[J]. arXiv preprint arXiv:2104.08320, 2021.

**Summary Of The Paper:**

This paper test 4 deep active learning mehods across 10 datasets (including text classification and multi-choice commonsense reasoning) based on pre-trained LMs. It conduct many ablation studies to explore whether some inherent factors of like batch size in active learning, the model size of pre-training models would affect the effectiveness of active learning with pre-trained LMs.

**Summary Of The Review:**

This paper conducted a large amount of empirical experiments on AL with pre-trained LMs. However, the research problems they explored are somewhat covered by existing research outputs and their conclusions are less attractive compared with [1]. Additionally, the writing of this article should also be improved.

---

> ### Author Response · Authors · 2021-11-23
> **Response to reviewer 4**
>
> Thank you for your review.  We hope the comments and revisions below will address your concerns:
>
> We believe that our work represents a significant contribution over previous work.  While our work does build on [1], we do extensive experiments on more tasks with the pruning technique proposed in it and find that it doesn't consistently explain AL's behavior.  We propose an alternative hypothesis (AL selects examples that increase instability, but are not strictly harmful) and find that it fits the observations at least as well as the collective outlier hypothesis from [1] in our experiments.  We would be interested to learn more about why the reviewer finds our comparison to [1] unsatisfying.
>
> Regarding Margatina et al [2], we agree that it would have been interesting to apply their techniques, but for a few reasons we would argue that it was reasonable not to do so (aside from the fact that it has not yet been peer-reviewed):
> - That work doesn't investigate scenarios where AL fails to outperform random (and therefore doesn’t identify our hypothesis of instability as a cause of such failures); for the datasets that were included in both their work and ours, our results qualitatively agree that AL improves over random.
> - They study the effects of TAPT on the absolute performance of AL methods but do not appear to state conclusions on its effects on the performance gap we study, between AL and random sampling.
>
> Regarding the choice of quota, we refer the reviewer to our general response.
>
> Finally, with respect to the writing concerns, the second contribution was not incomplete.  The intent of the trailing "and" was to indicate a list (contributions X, Y, "and" Z), but on re-reading we agree that it could be confusing and have revised it.

---

> > ### Comment · Reviewer_Seyb · 2021-11-30
> > **response**
> >
> > I have updated the score.

---

### Official Review · Reviewer_e7J6 · 2021-11-02

**Correctness:** 2
**Technical Novelty And Significance:** 1
**Empirical Novelty And Significance:** 2
**Recommendation:** 5
**Confidence:** 4

**Main Review:**

Strengths:

The authors connect active learning example selection to more severe training instability than random selection. Further, they show downside-variability (“greater instability on the losses than on the wins”) suggesting specific sets of examples are responsible for systemic failures in active learning, but not necessarily that other selections would greatly outperform the random baseline. Both of these observations are of interest to the NLP active learning community, adding depth to the characterization of the problem.

The wide range of datasets and active learning techniques they use (including BALD which prior works shows is very competitive) lends credence to the conclusions. It especially corroborates that certain tasks are innately more challenging for active learning than others.

Their thorough ablation experiments and other analysis yield some interesting findings the authors could emphasize more. For instance, the greater instability of larger Transformers to active learning bodes poorly for practitioners leveraging ever increasing model sizes for low-resource datasets. Additionally, the authors conclude that pre-training is not a significant factor in the efficacy of active learning, but their numerical results suggest active learning methods (Entropy and Coreset) narrow the gap with the random baseline significantly from BERT-Base to RoBERTa-Base! The margin of change seems even larger than some results which are discussed in the paper as significant. More discussion could bolster this finding.

Weaknesses:

The primary weakness of the paper is the lack of convincing justification that the authors have discovered a phenomenon distinct from “collective outliers” (Karamcheti et al., 2021) -- or if it is distinct, how exactly is it distinct?

1. For context, Karamcheti et al. (2021) introduce an "oracle" active learning method that prunes uninformative (outlier) examples from the unlabelled, selectable pool U, but it requires gold labels to filter these. This work tries a single pruning rate and finds it significantly degrades, rather than improves results: for both active learning and random selection. However, they only mention a single pruning rate of 50%, which could easily over-prune (all challenging as well as outlier examples) for the NLP datasets used here. Especially because of the surprising magnitude by which this pruning degrades absolute performance, it is unfortunately necessary to try more pruning rates for a fair comparison.

2. Even if pruning is ineffective (at multiple pruning rates), it’s never really explained why, despite this being a core contribution of the paper.

A qualitative analysis was required in Karamcheti et al., (2021) to explain “collective outliers”. A similar analysis here could greatly demystify why these sets of examples cause instability, and whether they are indeed “informative”.

There are a couple decisions related to methodology that may need some further examination.

1. The first is the total unlabelled pool size. It appears to vary by orders of magnitude according to Table 10 in the Appendix. While the batch selection L and total selection Q variables are constants, fixed at 25 and 500 respectively, the pool of examples to choose from varies wildly. This is very likely a confounding factor in the efficacy of active learning and pruning techniques. It deserves some discussion or control.

2. The second is the choice of L=25 examples chosen at each acquisition, and also as the seed set. I am concerned this is quite low and could yield exaggerated instability, especially for large Transformer models. If I read them correctly, Ein-Dor et al. (2020) use at least L=50 for simple classification tasks, and Karamcheti et al., (2021) use L>= 400 and larger seed sets. The authors investigate L=12 in the ablation, but in a real setting, it seems unlikely practitioners will label <50 examples before re-training.

Particularly because the reported accuracy margins are so slim, either of these variables could modify the empirical conclusions. A statistical test on the Average performances would also improve confidence in the conclusions.

Questions and minor suggestions for the authors:

1. Why is Entropy missing from Table 2? And BALD-MC from Tables 3 and 4? And why is BALD missing again from Table 6? Computational expense is used to justify the omission of BALD in one instance, but the others appear inconsistent?
2. It’s very hard to compare the absolute differences Tables 1 and 2 for ourselves. Perhaps there is some visual representation that could help demonstrate the comparisons you make in the text?
3. Margins are very small for the Average differences across all datasets as well -- have you considered confidence intervals on those as well?
4. Minor: Only half the datasets are shown in Tables 3 and 4, but it’s unclear how/why those were chosen?



**Summary Of The Paper:**

This work evaluates active learning methods for pre-trained Transformer models, and diagnoses their inconsistent performance as a consequence of training examples which exacerbate variability in test-time accuracy. They propose re-training the model repeatedly to circumvent bad outcomes due to variability, and show this works particularly well on larger pre-trained models. The authors conduct a wide range of experiments, over many datasets, and also a series of ablation studies to reinforce their claim that these active learning methods are ineffective for pre-trained Transformers.

**Summary Of The Review:**

In summary, we would encourage the authors to more rigorously examine the heart of their work -- the source of the instability they uncover: is pruning really ineffective, or are we removing too many challenging examples along with the outliers? Can we qualitatively diagnose the problem and explain what makes an example “informative” but unstable? And as accuracy margins are so small, it may be unfortunately necessary to justify or relax some hyperparameter choices, and run additional experiments to confirm these findings generalize.

---

> ### Author Response · Authors · 2021-11-23
> **Response to reviewer 3**
>
> Thank you for your review.  We have made several revisions to address your comments, as we describe below:
>
> **Distinction from collective outliers**
>
> We believe our instability hypothesis is distinct from but complementary to the collective outlier hypothesis.  If collective outliers were the sole cause of poor AL performance, we would not expect multi-training to help (because the examples are simply unlearnable), but we observe that it helps just as much as pruning does.  This does not rule out the possibility that collective outliers **also** contribute to AL’s behavior, but it does suggest that they are a distinct phenomenon from the instability we observe.  We have updated the text in the introduction and at the end of Section 3.3 to make this clearer.
>
> **Pool size and Q**
>
> We do not observe a significant relationship between the pool size and the benefit of AL in our experiments.  For example, CODAH (our third-smallest dataset) and SWAG (our third largest) both consistently fail to improve across all AL methods in Table 1, whereas AGN-SB (our second-smallest dataset) and AGN (our largest) both get a boost.
>
> As other reviews also commented on the choice of Q, we have included a discussion of that in our general response.
>
> **Pretraining ablation**
>
> The observation on pretraining is interesting; we had primarily been testing the opposite hypothesis (that pretraining could harm AL effectiveness), but you are correct that on several datasets AL appears to work differently on RoBERTa than BERT.  Coreset is a bit worse on BERT, and interestingly our newly-added BatchBALD experiments show the opposite effect, with BatchBALD doing much better on BERT than on RoBERTa.  We have updated the text with related discussion in section 3.4.3 (page 8) and adjusted our wording where we mention pretraining in the introduction and conclusion.
>
> **Other comments**
>
> - Time constraints prevented us from running all combinations of AL methods and datasets for all ablations, but we have now updated the paper with Entropy in Table 2 (page 4) and BatchBALD in Table 7 (page 8), and added a clarification in the table captions about BALD being omitted in some cases (in short, it is very expensive to run and generally a bit worse than BatchBALD).
>
> - We have added standard deviation numbers to the averages in all tables.

---

> > ### Comment · Reviewer_e7J6 · 2021-11-27
> > **Response to Authors**
> >
> > We thank the authors for their updates and new results. Filling in the missing methods in multiple tables, the standard deviations, additional experiments for budget size, and analysis of the pool size help address some of our concerns, and lend greater evidence towards their conclusions. These efforts are appreciated, and we will adjust our score upwards to reflect this.
> >
> > Of their three stated contributions in the introduction, the first and third now appear well substantiated, but a few questions remain dubious regarding the “novel explanation” for active learning instability, and, related, why pruning degrades results.
> > * If we’re not mistaken, their claim could be simplified to: (i) active learning selects harder examples, and (ii) fine-tuning large pre-trained Transformers, which is known to be unstable in general [1], is more unstable when harder examples are selected. This is a nice observation and would explain why multi-training is effective, but we aren’t sure that it is either novel or particularly surprising.
> > * Multi-training is presented both as a diagnosis and fix for “informative” but unstable examples selected by active learning. However, we are not sure the reported results (now with standard errors) reflect the authors’ conclusions. Table 5 shows Multi-Training improves mean Random selection performance from 68.30 (0.09) to 68.84 (0.07), while Multi-Training improves BatchBALD-MC mean performance from 68.32 (0.07) to 68.99 (0.1). This translates to a mean difference of +0.54 for random, or +0.67 for BatchBALD-MC, but the difference between these two is within the standard deviations.
> > * To support the claim that the authors have found an interesting or novel phenomena, they would need to demystify why pruning degrades performance here, and qualitatively compare examples, as mentioned in our first review, similar to Karamcheti et al. (2021). We appreciate the authors added a new pruning rate of 25%. This lower pruning rate significantly reduced the performance degradation initially seen at 50%, for both Random and BatchBALD-MC, suggesting many beneficial/challenging examples were indeed being removed (our original critique). The reader is left to wonder if there are simply fewer collective outliers in these NLP datasets than in VQA, and if this could be a sufficient explanation for instability and multi-training?
> >
> > In summary, the authors show that active learning rarely outperforms random selection for large pretrained transformers, and conduct interesting analyses on the effect of pre-training, model size, and adversarial filtering. Altogether, these are interesting findings, however without adequate support or analysis for one of their primary contributions, articulated as a “novel explanation” for active learning instability, is not yet clear or convincing that it merits a score higher than a 5.
> >
> > [1] Dodge, Jesse, et al. "Fine-tuning pretrained language models: Weight initializations, data orders, and early stopping." arXiv preprint arXiv:2002.06305 (2020).

---

> > > ### Author Response · Authors · 2021-11-30
> > > **Reply to reviewer**
> > >
> > > We thank the reviewer for their updated score and additional insights.
> > >
> > > - We agree with the reviewer's simplification of our claim, but disagree about the lack of novelty.  While the final result may be somewhat intuitive, to our knowledge it has thus far not been addressed in AL literature, and in fact prior work (Karamcheti et al) suggested a different explanation for AL's failures.  We therefore argue that these conclusions, however obvious they might seem in retrospect, are currently being overlooked by the community.
> > >
> > > - We find the reviewer's second point reasonable.  While we believe that our results suggest that multi-training helps, the evidence for this is not conclusive.  If accepted, we will amend our phrasing for this claim to acknowledge the lack of statistical significance, unless we are able to verify the significance through additional experiments.
> > >
> > > - If it were the case that NLP datasets simply have fewer collective outliers, we would expect that AL would not fail on them, or at least that pruning would quickly enable AL to outperform random (unless there is an additional and complementary factor such as instability, as we claim).  However, we see that pruning begins to degrade absolute performance before bringing AL above random on some datasets (e.g. aNLI, SWAG, PIQA).  Therefore, we do not believe our results could be fully explained by NLP datasets having fewer collective outliers.

---

### Official Review · Reviewer_dz3i · 2021-11-02

**Correctness:** 3
**Technical Novelty And Significance:** 2
**Empirical Novelty And Significance:** 2
**Recommendation:** 6
**Confidence:** 4

**Main Review:**

Strength:
- Experiments on many data sets.
- The idea of using convex hull of AUC is interesting, but it would it nice to confirm it experimentally.

Weakness:
- Q seems to be too small, which might amplify the instability issue.
- Little novelty

While their experiments, and findings are interesting, it leaves many questions that could (at least in principle) be answered by more experiments.
Their main finding seems to be that larger transformer models lead to more instability for training with the instances selected by AL, Table 4 and Table 5.
However, Table 4 is only an approximation to results that could be achieved with training with different initializations. Though computationally expensive it would be very interesting to see whether the results from the approximation can be confirmed.

Several questions:
- What exactly is meant by the Entropy method?
My understanding that is that Entropy is the same as Least Confidence, since high entropy <-> low confidence.
- Why is Entropy missing in Table 2?
- Why is Q set to 500? This seems to be too small. It would be interesting to learning curve graphs with larger Q.

**Summary Of The Paper:**

This work investigates Active Learning (AL) for fine-tuning large pre-trained transformer models (BERT, RoBERT) with applications to several NLP tasks.
They observe that depending on the task/dataset, AL actually does not improve over a random baseline.
Their ablation tests suggests that the instability of re-training the large models might be amplified by instances selected by AL. As a remedy, they suggest to train several times with different initializations, and then selecting the best model on the development data.

**Summary Of The Review:**

For an experimental paper, it is interesting, and might point to some new directions for AL.
However, more experiments to clarify some points (see above) should be added.

---

> ### Author Response · Authors · 2021-11-23
> **Response to reviewer 2**
>
> Thank you for your review.  We hope the comments below will answer your questions:
>
> - We also would have been interested to try multi-training with roberta-large to validate the convex hull approximation.  However, due to the use of both multi-training and a large model, this experiment is especially expensive.  If accepted, we will run it for one or two datasets in the final version.
>
> - Entropy is very similar to least confidence (evidenced by some preliminary experiments we did that found almost identical results) and they make identical decisions for binary classification tasks, but they are not precisely the same for non-binary classification tasks. For example, consider a 4-way classification problem for which the model predicts the distributions [0.1, 0.1, 0.1, 0.7] and [0, 0, 0.4, 0.6] for two examples in the pool.  Their entropies are respectively 1.36 and 0.97 bits but their confidences are 0.7 and 0.6.  So the first would be selected by maximum entropy while the second one would be selected by least confidence.
>
> - Time constraints prevented us from running all combinations of AL methods and datasets for all ablations, but we have now updated the paper with Entropy in Table 2 (page 4) and BatchBALD in Table 7 (page 8).
>
> - Please see our overall response regarding the setting of Q.

---

> > ### Comment · Reviewer_dz3i · 2021-11-24
> > **thank you**
> >
> > thank you for your detailed explanations.
> > As future work it might be interesting to study how the instability issue is affected by the size of the initial training data size.
> > Currently the initial training data size is equal to the batch size, making it larger, the instability is expected to be reduced, but it is unclear by how much.

---

### Official Review · Reviewer_igJc · 2021-11-03

**Correctness:** 4
**Technical Novelty And Significance:** 3
**Empirical Novelty And Significance:** 2
**Recommendation:** 6
**Confidence:** 3

**Main Review:**

Strengths:
 - I could learn various empirical observations of how active learning works in these large pre-trained models on various NLP tasks. I hope to see this paper at the conference and others can learn from it as well.

Weakness:
 - It would be nice to see some macro analysis of data points showing the overall tendency of each setup.
 - I hope to see more insights regarding how these two different AL methods (density-based and entropy-based) would work in a different way.

**Summary Of The Paper:**

This paper examines how active learning works with large pre-trained models, with a novel explanation with empirical evidence.

**Summary Of The Review:**

Good empirical findings from extensive experiments, but limited to the individual setup in specific conditions without an overall picture of dataset dynamics.

---

> ### Author Response · Authors · 2021-11-23
> **Response to reviewer 1**
>
> Thank you for your review.
>
> Regarding your question about the kinds of selections AL makes, Reviewer 5 asked a similar question so we provide the same response here:
> From our exploratory analysis we have observed that Coreset heavily favors longer examples, and generally Entropy does as well to a far lesser extent. Analysis of other qualitative patterns among selected examples is still an item for future work, as thus far we have not observed any clear trend.

---

### Author Response · Authors · 2021-11-23
**General response to reviewers**

We thank all of the reviewers for their time and insights.  We have made several revisions to the paper to clarify some points and added several experiments requested by reviewers:

- Entropy results with pruning in Table 2 (page 4)
- BatchBALD results for BERT in Table 7 (page 8)
- BatchBALD results with a 25% pruning threshold in Table 11 in Appendix C (page 15; also includes results from Tables 1 and 2 side-by-side for easier comparison)
- BatchBALD results with a larger budget and batch size in Table 12 in Appendix D (page 16)

For convenience, the differences between our new submission and the original submission can be found in the file `diff.pdf` in the supplementary material.

We also reiterate our main contributions here for reference:
- On a wide range of methods and datasets, we find that AL does not consistently improve large LMs.
- We propose a **novel explanation** for the failures, namely that AL has the effect of increasing instability on larger models.
- Through a series of ablations, we find that acquisition batch size and adversarial filtering do not explain the poor performance, while pretraining has inconsistent effects between AL methods

Several reviewers have commented on our choice of annotation budget Q.  While the choice of Q does impact the behavior of AL (demonstrated in a newly-added ablation in the appendix), we argue that our selection is reasonable, as in a real-world setting we may have a limited budget to spend on annotation and not be able to collect more than a small number of examples (thus motivating the desire to use AL to get the best performance at that budget).

Similarly, the choice of batch size |ΔL| may have an effect, but our batch size ablation didn't suggest that the effect is substantial.  Theoretically a smaller |ΔL| is preferred as it reduces the likelihood of selecting correlated examples within a batch, and many AL works consider the scenario of online AL, where essentially |ΔL| is set to 1.

---

### Decision · Program_Chairs · 2022-01-20

**Decision:**

Reject

**Comment:**

The paper proposes a novel explanation for the ineffectiveness of active learning (AL), namely, that AL will select unlearnable collective outliers.

Reviewers generally find the finding is interesting, but the paper lacks in-depth analyses. There're additional concerns on the experimental setups.